# FleXOR: Trainable Fractional Quantization

**Dongsoo Lee**[*]  **Se Jung Kwon**[*]  **Byeongwook Kim**
**Yongkweon Jeon**  **Baeseong Park**  **Jeongin Yun**
Samsung Research, Seoul, Republic of Korea
{dongsoo3.lee, sejung0.kwon, byeonguk.kim,
dragwon.jeon, bpbs.park, ji6373.yun}@samsung.com

## Abstract

Quantization based on the binary codes is gaining attention because each quantized bit can be directly utilized for computations without dequantization using look-up tables. Previous attempts, however, only allow for integer numbers of quantization bits, which ends up restricting the search space for compression ratio and accuracy. In this paper, we propose an encryption algorithm/architecture to compress quantized weights so as to achieve fractional numbers of bits per weight. Decryption during inference is implemented by digital XOR-gate networks added into the neural network model while XOR gates are described by utilizing $\tanh(x)$ for backward propagation to enable gradient calculations. We perform experiments using MNIST, CIFAR-10, and ImageNet to show that inserting XOR gates learns quantization/encrypted bit decisions through training and obtains high accuracy even for fractional sub 1-bit weights. As a result, our proposed method yields smaller size and higher model accuracy compared to binary neural networks.

## 1   Introduction

Deep Neural Networks (DNNs) demand a larger number of parameters and more computations to support various task descriptions all while adhering to ever-increasing model accuracy requirements. Because of abundant redundancy in DNN models [9, 5, 3], numerous model compression techniques are being studied to expedite the inference of DNNs [19, 16]. As a practical model compression scheme, parameter quantization is a popular choice because of the high compression ratio and regular formats after compression so as to enable full memory bandwidth utilization.

Quantization schemes based on binary codes are gaining increasing attention since quantized weights follow specific constraints to allow simpler computations during inference. Specifically, using the binary codes, a weight vector is represented as $\sum_{i=1}^{q}(\alpha_i \boldsymbol{b}_i)$, where $q$ is the number of quantization bits, $\alpha$ is a scaling factor ($\alpha \in \mathbb{R}$), and each element of a vector $\boldsymbol{b}_i$ is a binary $\in \{-1, +1\}$. Then, a dot product with activations is conducted as $\sum_{i=1}^{q}(\alpha_i \sum_{j=1}^{v} a_j b_{i,j})$, where $a_j$ is a full-precision activation and $v$ is the vector size. Note that the number of multiplications is reduced from $v$ to $q$ (expensive floating-point multipliers are less required for inference). Moreover, even though we do not discuss a new activation quantization method in this paper, if activations are also quantized by using binary codes, then most computations are replaced with bit-wise operations (using XNOR logic and population counts) [25, 20]. Consequently, even though representation space is constrained compared with quantization methods based on look-up tables, various inference accelerators can be designed to exploit the advantages of binary codes [20, 25]. Since a successful 1-bit weight quantization method has been demonstrated in BinaryConnect [3], advances in compression-aware training algorithms in the form of binary codes (e.g., binary weight networks [20] and LQ-Nets [27]) produce 1-3 bits for quantization while accuracy drop is modest or negligible. Fundamental

---

[*]Equal Contribution.

| | Weights on MEM | Converter | Internal Weight Format | Computation Scheme |
|---|---|---|---|---|
| **Ours (FleXOR)** | **Encrypted *W*** *(w/ Scaling Factors)* | **XOR Decryptor** | **Binarized *W*** | **Bit-wise Computation** |
| **Binary-coding-based Quantization** | **Binarized *W*** *(w/ Scaling Factors)* | | **Binarized *W*** | **Bit-wise Computation** |
| **Vector Quantization** | **Quantized *W*** *(w/ Codebooks)* | **Looking up Codebooks** | ***W*** **(Full Precision)** | **Full Precision Computation** |

Figure 1: Dataflow and computation formats of binary-coding-based quantization, vector quantization, and our proposed quantization scheme.

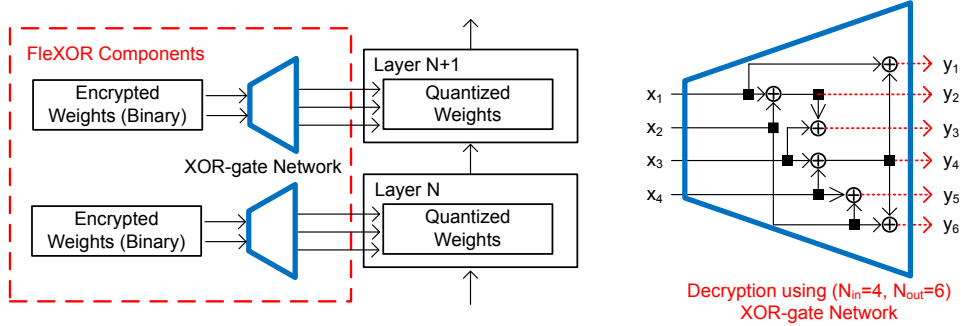

Figure 2: FleXOR components added to the quantized DNNs to compress quantized weights through encryption. Encrypted weight bits are decrypted by XOR gates to produce quantized weight bits.

investigations on DNN training mechanisms using fewer quantization bits have also been actively reported [17, 2].

Previously, binary-coding-based quantization has only permitted integer numbers of quantization bits, limiting the compression/accuracy trade-off search space, especially in the range of very low quantization bits. In this paper, we propose a flexible encryption algorithm/architecture (called "FleXOR") to enable fractional sub 1-bit numbers to represent each weight while quantized bits are trained by gradient descent. Even though vector quantization is also a well-known scheme with a high compression ratio [22], we assume the form of binary codes. Note that the number of quantization bits can be different for each layer (e.g., [24]) to allow fractional quantization bits on average. FleXOR implies fractional quantization bits for each layer that can be quantized with a different number of bits.

To the best of our knowledge, **our work is the first to explore model accuracy under 1 bit/weight when weights are quantized based on the binary codes.** Figure 1 compares representations of weights to be stored in memory, converting method, and computation schemes of three quantization schemes. FleXOR maintains the advantages of binary-coding-based quantization (i.e., dequantization is not necessary for the computations) while quantized weights are further compressed by encryption. Note that since our major contribution is to enable fractional sub 1-bit weight quantization, for experiments, we selected models that have been previously quantized by '1' bit/weight for comparisons on the model accuracy. As a result, unfortunately, the range of model selections is somewhat limited.

## 2 Encrypting Quantized Bits using XOR Gates

The main purpose of FleXOR is to compress quantized bits into encrypted bits that can be reconstructed by XOR gates as shown in Figure 2. Suppose that $N_{out}$ bits are to be compressed into $N_{in}$ bits ($N_{out} > N_{in}$). The role of an XOR-gate network is to produce various $N_{out}$-bit combinations using $N_{in}$ bits [14]. In other words, in order to maximize the chance of generating a desirable set of quantized bits, the encryption scheme is designed to seek a particular property where all possible $2^{N_{in}}$ outcomes through decryption are evenly distributed in $2^{N_{out}}$ space.

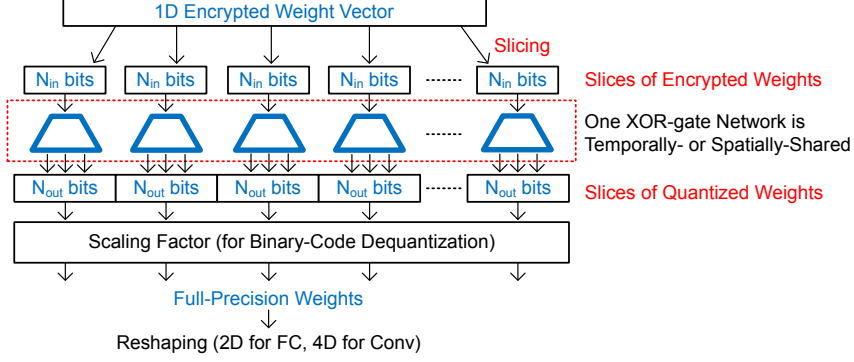

Figure 3: Encrypted weight bits are sliced and reconstructed by a XOR-gate network which can be shared (in time or space). Then quantized bits after XOR gates are finally reshaped.

A linear Boolean function, $f(\boldsymbol{x})$, maps $f : \{0,1\}^{N_{in}} \to \{0,1\}$ and has the form of $a_1 x_1 \oplus a_2 x_2 \oplus \cdots \oplus a_{N_{in}} x_{N_{in}}$ where $a_j \in \{0,1\}$ ($1 \le j \le N_{in}$) and $\oplus$ indicates bit-wise modulo-2 addition. In Figure 2, six binary outputs are generated through six Boolean functions using four binary inputs. Let $f_1(\boldsymbol{x})$ and $f_2(\boldsymbol{x})$ be two such linear Boolean functions using $\boldsymbol{x} = (x_1, x_2, ..., x_{N_{in}}) \in \{0,1\}^{N_{in}}$. The Hamming distance between $f_1(\boldsymbol{x})$ and $f_2(\boldsymbol{x})$ is the number of inputs on which $f_1(\boldsymbol{x})$ and $f_2(\boldsymbol{x})$ differ, and defined as

$$d_H(f_1, f_2) := w_H(f_1 \oplus f_2) = \#\{\boldsymbol{x} \in \{0,1\}^{N_{in}} | f_1(\boldsymbol{x}) \ne f_2(\boldsymbol{x})\}, \qquad (1)$$

where $w_H(f) = \#\{\boldsymbol{x} \in \{0,1\}^{N_{in}} | f(\boldsymbol{x}) = 1\}$ is the Hamming weight of a function and $\#\{\}$ corresponds to the size of a set [12]. The Hamming distance is a well-known method to express non-linearity between two Boolean functions [12] and increased Hamming distance between a pair of two Boolean functions results in a variety of outputs produced by XOR gates. Increasing Hamming distance is a required feature for cryptography to derive complicated encryption structure such that inverting encrypted data becomes difficult. For digital communication, the Hamming distance between encoded signals is closely related to the amount of error correction possible.

FleXOR should be able to select the best out of $2^{N_{in}}$ possible outputs that are randomly selected from larger $2^{N_{out}}$ search space. Encryption performance of XOR gates is determined by the randomness of $2^{N_{in}}$ output candidates, and is enhanced by increasing Hamming distance that is achieved by larger $N_{out}$ (for a fixed compression ratio). Now, let $N_{tap}$ be the number of 1's in a row of $\boldsymbol{M}^{\oplus}$. Another method to enhance encryption performance is to increase $N_{tap}$ so as to increase the number of shuffles (through more XOR operations) using encrypted bits to generate quantized bits such that correlation between quantized bits is reduced.

In Figure 2, $y_1$ is represented as $x_1 \oplus x_3 \oplus x_4$, or equivalently a vector $[1\,0\,1\,1]$ denoting which inputs are selected. Concatenating such vectors, a XOR-gate network in Figure 2 can be described as a binary matrix $\boldsymbol{M}^{\oplus} \in \{0,1\}^{N_{out} \times N_{in}}$ (e.g., the second row of $\boldsymbol{M}^{\oplus}$ is $[1\,1\,0\,0]$ and the third row is $[1\,1\,1\,0]$). Then, decryption through XOR gates is simply represented as $\boldsymbol{y} = \boldsymbol{M}^{\oplus}\boldsymbol{x}$ where $\boldsymbol{x}$ and $\boldsymbol{y}$ are the binary inputs and binary outputs of XOR gates, and addition is 'XOR' and multiplication is 'AND' (see Appendix for more details and examples).

Encrypted weight bits are stored in a 1-dimensional vector format and sliced into blocks of $N_{in}$-bit size as shown in Figure 3. Then, the decryption of each slice is performed by an XOR-gate network that is shared by all slices (temporally- or spatially-shared). Depending on the quantization scheme and characteristics of layers, quantized bits may need to be scaled by a scaling factor and/or reshaped. Area and latency overhead induced by XOR gates are negligible as demonstrated in VLSI testing and parameter pruning works [23, 16, 1].

Since an XOR-gate network is shared by many weights (i.e., $\boldsymbol{M}$ is fixed for all slices), it is difficult (if not impossible) to manually optimize an XOR-gate network. Hence, a random $\boldsymbol{M}$ configuration is enough to fulfill the purpose of random number generation. In short, the XOR-gate network design is simple and straightforward.

## 3 FleXOR Training Algorithm for Quantization Bits Decision

Once the structure of XOR gates has been pre-determined and fixed to increase the Hamming distance of XOR outputs, we find quantized and encrypted bits by adding XOR gates into the model. In other words, we want an optimizer that understands the XOR-gate network structure so as to compute encrypted bits and scaling factors via gradient descent. For inference, we store binary encrypted weights (converted from real number encrypted weights) in memory and generate binary quantized weights through Boolean XOR operations. Activation quantization is not discussed in this paper to avoid the cases where the choice of activation quantization method affects the model accuracy.

Similar to the STE method introduced in [3], Boolean functions need to be described in a differentiable manner to obtain gradients in backward propagation. For two real number inputs $x_1$ and $x_2$ ($x_1, x_2 \in \mathbb{R}$ to be used as encrypted weights), the Boolean version of a XOR gate for forward propagation is described as (note that $0$ is replaced with $-1$)

$$\mathcal{F}^{\oplus}(x_1, x_2) = (-1)\,\text{sign}(x_1)\,\text{sign}(x_2). \tag{2}$$

For inference, we store $\text{sign}(x_1)$ and $\text{sign}(x_2)$ instead of $x_1$ and $x_2$. On the other hand, a differentiable XOR gate for backward propagation is presented as

$$f^{\oplus}(x_1, x_2) = (-1)\tanh(x_1 \cdot S_{\text{tanh}})\tanh(x_2 \cdot S_{\text{tanh}}), \tag{3}$$

where $S_{\text{tanh}}$ is a scaling factor for FleXOR. Note that $\tanh$ functions are widely used to approximate Heaviside step functions (i.e., $y(x){=}1$ if $x{>}0$ or $0$, otherwise) in digital signal processing and $S_{\text{tanh}}$ can control the steepness. In [6, 15], $\tanh$ is also suggested to approximate the STE function. In our work, on the other hand, $\tanh$ is to proposed to make XOR operations trainable for 'encryption.' rather than 'quantization.' In the case of consecutive XOR operations, the order of inputs to be fed into XOR gates should not affect the computation of partial gradients for XOR inputs. Therefore, as a simple extension of Eq. (3), a differentiable XOR gate network with $n$ inputs can be described as

$$f^{\oplus}(x_1, x_2, \ldots, x_n) = (-1)^{n-1}\tanh(x_1 \cdot S_{\text{tanh}})\tanh(x_2 \cdot S_{\text{tanh}})\ldots\tanh(x_n \cdot S_{\text{tanh}}). \tag{4}$$

Then, a partial derivative of $f^{\oplus}$ with respect to $x_i$ (an encrypted weight) is given as

$$\frac{\partial f^{\oplus}(x_1, x_2, \ldots, x_n)}{\partial x_i} = S_{\text{tanh}}(-1)^{n-1}(1 - \tanh^2(x_i \cdot S_{\text{tanh}}))\frac{\prod_{j=1}^{n}\tanh(x_j \cdot S_{\text{tanh}})}{\tanh(x_i \cdot S_{\text{tanh}})} \tag{5}$$

Note that increasing $N_{tap}$ is associated with more $\tanh$ multiplications for each XOR-gate network output. From Eq. (5), thus, increasing $N_{tap}$ may lead to the vanishing gradient problem since $|\tanh(x)| \leq 1$. To resolve this problem, we also consider a simplified alternative partial derivative expressed as

$$\frac{\partial f^{\oplus}(x_1, x_2, \ldots, x_n)}{\partial x_i} \approx S_{\text{tanh}}(-1)^{n-1}(1 - \tanh^2(x_i \cdot S_{\text{tanh}}))\prod_{j \neq i}\text{sign}(x_j). \tag{6}$$

Compared to Eq. (5), approximation in Eq. (6) is obtained by replacing $\tanh(x \cdot S_{\text{tanh}})$ with $\text{sign}(x)$. Eq. (6) shows that when we compute a partial derivative, all XOR inputs other than $x_i$ are assumed to be binary, i.e., the magnitude of a partial derivative is then only determined by $x_i$. We use Eq. (6) in this paper to calculate custom gradients of encrypted weights due to fast training computations and convergence, and use Eq. (2) for forward propagation.

By training the whole network including FleXOR components using custom gradient computation methods described above, encrypted and quantized weights are obtained in a holistic manner. FleXOR operations for convolutional layers are described in Algorithm 1, where encrypted weights (inputs of an XOR-gate network) and quantized weights (outputs of an XOR-gate network) are $\boldsymbol{w}^e$ and $\mathbf{W}^q$. We note that Algorithm 1 describes hardware operations (that are best implemented by ASIC or FPGA) rather than instructions to be operated by CPUs or GPUs.

We first verify the basic training principles of FleXOR using LeNet-5 on the MNIST dataset. LeNet-5 consists of two convolutional layers and two fully-connected layers (specifically, 32C5-MP2-64C5-MP2-512FC-10SoftMax), and each layer is accompanied by an XOR-gate network with $N_{in}$ binary inputs and $N_{out}$ binary outputs. The quantization scheme follows a 1-bit binary code with full-precision scaling factors that are shared across weights for the same output channel number (for conv layers) or output neurons (for FC layers). Encrypted weights are randomly initialized with

**Algorithm 1:** Pseudo codes of Conv Layer with FleXOR when the kernel size is $k \times k$, the number of input channel and output channel are $C_{in}$ and $C_{out}$, respectively.

---

$\boldsymbol{w}^e \in R^{\lceil (k \cdot k \cdot C_{in} \cdot C_{out})/N_{out} \rceil \cdot N_{in}}$ &emsp;&emsp;&emsp;&emsp;&emsp;&emsp;&emsp; ▷ Encrypted weights

$\boldsymbol{M}^{\oplus} \in \{0, 1\}^{N_{out} \times N_{in}}$ &emsp;&emsp;&emsp;&emsp;&emsp;&emsp;&emsp;&emsp;&emsp; ▷ XOR gates (shared)

$\boldsymbol{\alpha} \in \mathbb{R}^{C_{out}}$ &emsp;&emsp;&emsp;&emsp;&emsp;&emsp;&emsp;&emsp; ▷ Scaling factors for each output channel

---

**Function** `FleXOR_Conv`(*input, stride, padding*):

&emsp;**for** $i \leftarrow 0$ **to** $\lceil (k \cdot k \cdot C_{in} \cdot C_{out})/N_{out} \rceil - 1$ **do**

&emsp;&emsp;**for** $j \leftarrow 1$ **to** $N_{out}$ **do**

&emsp;&emsp;&emsp;$w^q_{i \cdot N_{out}+j} \leftarrow (-1) \cdot \left( \prod_{l=1, M^{\oplus}_{j,l}=1}^{N_{in}} \mathtt{Sign^c}\left(w^e_{i \cdot N_{in}+l}\right) \cdot (-1) \right)$ &emsp;&emsp; ▷ Eq. (2)

&emsp;$\mathbf{W}^q \leftarrow$ `Reshape`$(\boldsymbol{w}^q, [k, k, C_{in}, C_{out}])$

&emsp;**return** `Conv`(*input*, $\mathbf{W}^q$, $\boldsymbol{\alpha}$, *stride, padding*) &emsp;&emsp;&emsp;&emsp; ▷ Conv. operation for binary codes

**Forward Function** `Sign^c`(*x*):

&emsp;**return** $\text{sign}(x)$

**Gradient Function** `Sign^c` *(x, $\nabla$)*:

&emsp;**return** $\nabla \cdot (1 - \tanh^2(x \cdot S_{\tanh})) \cdot S_{\tanh}$ &emsp;&emsp;&emsp;&emsp;&emsp;&emsp; ▷ Eq. (6)

---

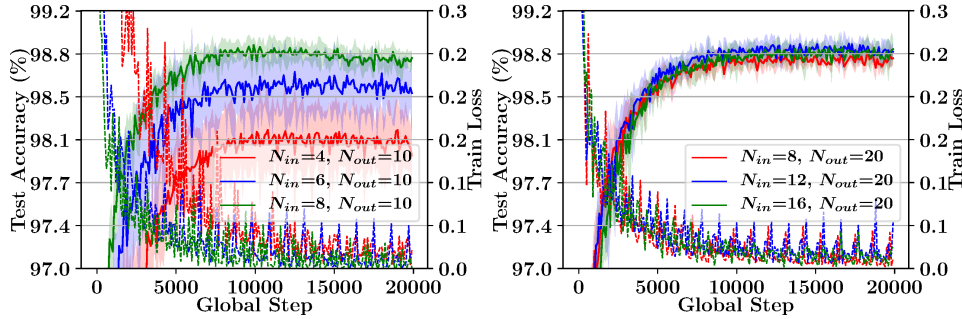

Figure 4: Test accuracy and training loss (average of 6 runs) with LeNet-5 on MNIST when $\boldsymbol{M}^{\oplus}$ is randomly filled with $\{0, 1\}$. $N_{out}$ is 10 or 20 to generate 0.4, 0.6, or 0.8 bit/weight quantization.

$\mathcal{N}(\mu{=}0, \sigma^2{=}0.001^2)$. All scaling factors for quantization are initialized to be 0.2 (note that if batch normalization layers are immediately followed, then scaling factors for quantization are redundant).

Using the Adam optimizer with an initial learning rate of $10^{-4}$ and batch size of 50 without dropout, Figure 4 shows training loss and test accuracy when $S_{\tanh}{=}100$, elements of $\boldsymbol{M}^{\oplus}$ are randomly filled with 1 or 0, and for two values of $N_{out}$ – 10 and 20. Using the 1-bit internal quantization method and $(N_{in}, N_{out})$ encryption scheme, one weight can be represented by $(N_{in}/N_{out})$ bits. Hence, Figure 4 represents training results for 0.4, 0.6, and 0.8 bits per weight. Note that as for a randomly filled $\boldsymbol{M}^{\oplus}$, increasing $N_{out}$ (and $N_{in}$ is determined correspondingly for the same compression ratio) increases the Hamming distance for a pair of any two rows of $\boldsymbol{M}^{\oplus}$ and, hence, offers the chance to produce more diversified outputs. Indeed, as shown in Figure 4, the results for $N_{out}{=}20$ present improved test accuracy and less variation compared with $N_{out}{=}10$. See Appendix for the distribution of encrypted weights at different training steps.

## 4 Practical FleXOR Training Techniques

In this section, we present practical training techniques for FleXOR using ResNet-32 [10] on the CIFAR-10 dataset [13]. We show compression results for ResNet-32 using fractional numbers as effective quantization bits, such as 0.4 and 1.2, that have not been available previously.

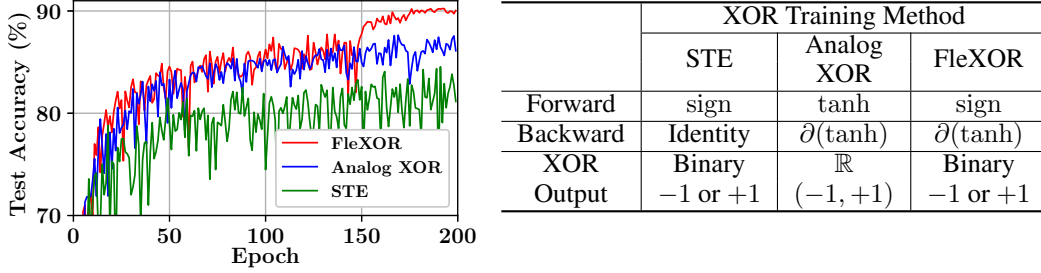

| | XOR Training Method | | |
|---|---|---|---|
| | STE | Analog XOR | FleXOR |
| Forward | sign | tanh | sign |
| Backward | Identity | $\partial(\tanh)$ | $\partial(\tanh)$ |
| XOR | Binary | $\mathbb{R}$ | Binary |
| Output | $-1$ or $+1$ | $(-1, +1)$ | $-1$ or $+1$ |

Figure 5: Test accuracy comparison on ResNet-32 (for CIFAR-10) using various XOR training methods. $N_{out}$=10, $N_{in}$=8, $q$=1 (thus, 0.8bit/weight), and $S_{\tanh}$=10.

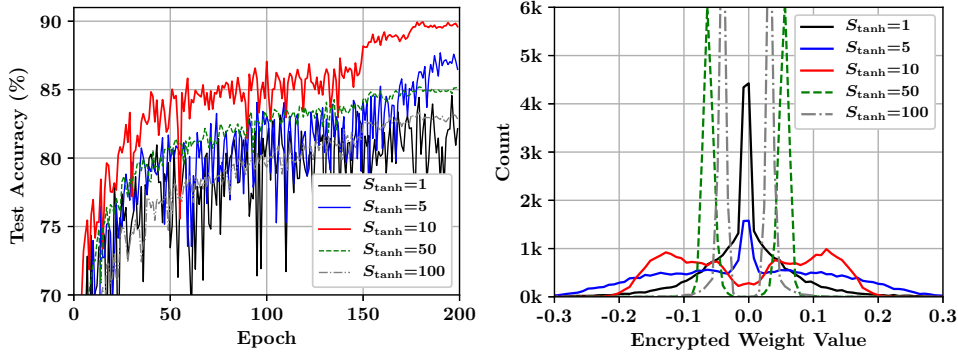

Figure 6: Test accuracy and distribution of encrypted weights (at the end of training) of ResNet-32 on CIFAR-10 using various $S_{\tanh}$ and the same $N_{out}$, $N_{in}$, and $q$ as Figure 5.

All layers, except the first and the last layers, are followed by FleXOR components sharing the same $\boldsymbol{M}^{\oplus}$ structure (thus, storage footprint of $\boldsymbol{M}^{\oplus}$ is ignorable). SGD optimizer is used with a momentum of 0.9 and a weight decay factor of $10^{-5}$. Initial learning rate is 0.1, which is decayed by 0.5 at the 150$^{\text{th}}$ and 175$^{\text{th}}$ epoch. As learning rate decays, $S_{\tanh}$ is empirically multiplied by 2 to cancel out the effects of weight decay on encrypted weights. The batch size is 128 and initial scaling factors of $\alpha$ are 0.2. '$q$' is the number of bits to represent binary codes for quantization. We provide some useful training insights below with relevant experimental results.

1) **Use small $N_{tap}$ (such as 2):** Large $N_{tap}$ can induce vanishing gradient problems in Eq. (5) or increased approximation error in Eq. (6). In practice, hence, FleXOR training with small $N_{tap}$ converges well with high test accuracy. Studying a training algorithm to understand a complex XOR-gate network with large $N_{tap}$ would be an interesting research topic that is beyond the scope of this work. Subsequently, we show experimental results using $N_{tap}$=2 in the remainder of this paper.

2) **Use 'tanh' rather than STE for XOR:** Since forward propagation for an XOR gate only needs a sign function, the STE method is also applicable to XOR-gate gradient calculations. Another alternative method to model an XOR gate is to use Eq. (3) for both forward and backward propagation as if the XOR is modeled in an analog manner (then, real number XOR outputs are quantized through STE). We compare three different XOR modeling schemes in Figure 5 with test accuracy measured when encrypted weights and XOR gates are converted to be binary for inference. FleXOR training method shows the best result because a) sign function for forward propagation enables estimating the impact of binary XOR computations on the loss function and b) $\partial(\tanh)$ for backward propagation approximates the Heaviside step function better compared to STE. Note that limited gradients from the tanh function eliminate the need for weight clipping, which is often required for quantization-aware training schemes [3, 25].

3) **Optimize $S_{\tanh}$:** $S_{\tanh}$ controls the smoothness of the tanh function for near-zero inputs. Large $S_{\tanh}$ employs large gradient for small inputs and, hence, results in well-clustered encrypted weight values as shown in Figure 6. Too large of a $S_{\tanh}$, however, hinders encrypted weights from being finely-tuned through training. For FleXOR, $S_{\tanh}$ is a hyper-parameter to be optimized empirically.

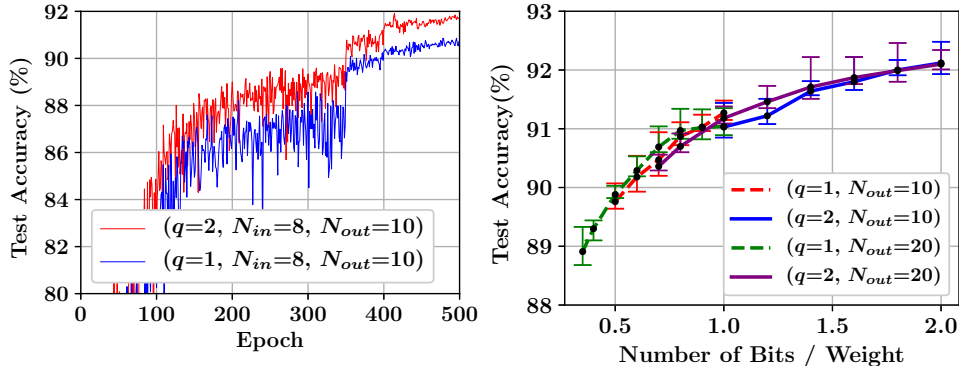

Figure 7: Test accuracy of ResNet-32 on CIFAR10 using learning rate warmup and various $q$, $N_{in}$, and $N_{out}$. The results on the right side are obtained by 5 runs.

Table 1: Weight compression comparison of ResNet-20 and ResNet-32 on CIFAR-10. For FleXOR, we use warmup scheme, $S_{\tanh}$=10, and $N_{out}$=20.

|  | ResNet-20 | | | ResNet-32 | | |
|---|---|---|---|---|---|---|
|  | FP | Compressed | Diff. | FP | Compressed | Diff. |
| BWN (1 bit) | 92.68% | 87.44% | -5.24 | 93.40% | 89.49% | -4.51 |
| BinaryRelax (1 bit) | 92.68% | 87.82% | -4.86 | 93.40% | 90.65% | -2.80 |
| LQ-Net (1 bit) | 92.10% | 90.10% | -1.90 | - | - | - |
| DSQ (1 bit) | 90.70% | 90.24% | -0.56 | - | - | - |
| FleXOR (1.0 bit) |  | 90.44% | -1.47 |  | 91.36% | -0.97 |
| FleXOR (0.8 bit) | 91.87% | 89.91% | -1.90 | 92.33% | 91.20% | -1.13 |
| FleXOR (0.6 bit) |  | 89.16% | -2.71 |  | 90.43% | -1.90 |
| FleXOR (0.4 bit) |  | 88.23% | -3.64 |  | 89.61% | -2.72 |

4) **Learning rate and $S_{\tanh}$ warmup:** Learning rate starts from 0 and linearly increases to reach the initial learning rate at a certain epoch as a warmup. Learning rate warmup is a heuristic scheme, but widely accepted to improve generalization capability mainly by avoiding large learning rate in the initial phase [11, 8]. Similarly, $S_{\tanh}$ starts from 5, and linearly increases to 10 using the same warmup schedule of the learning rate.

5) **Try various $q$, $N_{in}$, and $N_{out}$:** Using a warmup scheme for 100 epochs and learning rate decay by 50% at the $350^{th}$, $400^{th}$, and $450^{th}$ epoch, Figure 7 presents test accuracy of ResNet-32 with various $q$, $N_{in}$, and $N_{out}$. For $q>1$, different $M^{\oplus}$ configurations are constructed and then shared across all layers. Note that even for the 0.4bit/weight configuration (using $q$=1, $N_{in}$=8, and $N_{out}$=20), high accuracy close to 89% is achieved. 0.8bit/weight can be achieved by two different configurations (as shown on the right side of Figure 7) using ($q$=1, $N_{in}$=8, $N_{out}$=10) or ($q$=2, $N_{in}$=8, $N_{out}$=20). Interestingly, those two configurations show almost the same test accuracy, which implies that FleXOR is able to provide a linear relationship between the number of encrypted weights and model accuracy (regardless of internal configurations). In general, lowering $q$ reduces the number of computations with quantized weights.

We compare quantization results of ResNet-20 and ResNet-32 on CIFAR-10 using different compression schemes in Table 1 (with full-precision activation). BWN [20], BinaryRelax [26], and LQ-Net [27] propose different training algorithms for the same quantization scheme (i.e., binary codes). The main idea of these methods is to minimize quantization error and to obtain gradients from full-precision weights while the loss function is aware of quantization. Because all of the quantization schemes in Table 1 uses $q$=1 and binary codes, the amount of computations using quantized weights is the same. FleXOR, however, allows reduced memory footprint and bandwidth which are critical for energy-efficient inference designs [9, 1].

Note that even though achieving the best accuracy for 1.0 bit/weight is not the main purpose of FleXOR (e.g., XOR gate may be redundant for $N_{in}$=$N_{out}$), FleXOR shows the minimum accuracy drop for ResNet-20 and ResNet-32 as shown in Table 1. It would be an exciting research topic to

Table 2: ResNet-20 quantized by FleXOR with various $M^{\oplus}$ assigned to layers ($N_{out}$=20 for all layers). We divide 20 layers into three groups of layers except for the first and last layers.

| $N_{in}$ (Bits/Weight) | | | Average Bits/Weight | Accuracy |
|---|---|---|---|---|
| Layer 2–7 (13.5k params) | Layer 8–13 (45k params) | Layer 14–19 (180k params) | | |
| Fixed to be 12 (0.60) | | | 0.60 | 89.16% |
| 19 (0.95) | 19 (0.95) | 8 (0.40) | 0.53 | 89.23% (+0.07) |
| 16 (0.80) | 16 (0.80) | 8 (0.40) | 0.50 | 89.19% (+0.03) |
| 19 (0.95) | 16 (0.80) | 7 (0.35) | **0.47** | **89.29% (+0.13)** |

Table 3: Weight compression comparison of ResNet-18 on ImageNet.

| Methods | Bits/Weight | Top-1 | Top-5 | Storage Saving |
|---|---|---|---|---|
| Full Precision [10] | 32 | 69.6% | 89.2% | $1\times$ |
| BWN [20] | 1 | 60.8% | 83.0% | $\sim 32\times$ |
| ABC-Net [18] | 1 | 62.8% | 84.4% | $\sim 32\times$ |
| BinaryRelax [26] | 1 | 63.2% | 85.1% | $\sim 32\times$ |
| DSQ [7] | 1 | 63.7% | - | $\sim 32\times$ |
| FleXOR ($N_{out} = 20$) | 0.8 | 63.8% | 84.8% | $\sim 40\times$ |
| | 0.63 (mixed)[2] | 63.3% | 84.5% | $\sim 50.8\times$ |
| | 0.6 | 62.0% | 83.7% | $\sim 53\times$ |

study the distribution of the optimal number of quantization bits for each weight. We believe that such distribution would be wide and some weights require >1b while numerous weights need <1b because 1) increasing $N_{in}$ and $N_{out}$ allows such distributions to be wider and enhances model accuracy even for the same compression ratio and 2) as shown in Table 1, model accuracy of 1-bit quantization with FleXOR is higher than other quantization schemes that do not include encoding schemes.

While binary neural networks allow only 1-bit quantization as the minimum, FleXOR can assign any fractional quantization bits (less than 1) to different layers. Such a property is especially useful when some layers exhibit high redundancy and relatively less importance such that very low number of quantization bits do not degrade accuracy noticeably [4, 24]. To demonstrate mixed-precision quantization (with all less than 1-bit) enabled by FleXOR, we conduct experiments with ResNet-20 on CIFAR-10 while employing three different XOR-gate structures (i.e. multiple configurations of $M^{\oplus}$ are provided to different layer groups.). Table 2 shows that FleXOR with differently optimized $M^{\oplus}$ for each layer group can achieve a higher compression ratio with a smaller storage footprint compared to the case of FleXOR associated with just one common $M^{\oplus}$ configuration for all layers. When $N_{out}$ is fixed to be 20 for all layers, due to the varied importance of each group, small $N_{in}$ is allowed for the third group (of layers with a large number of parameters) while relatively large $N_{in}$ is selected for small layers. Compared to the case of $N_{in}$=12 for all layers (with 0.6 bits/weights), adaptively chosen $N_{in}$ sets (i.e., 19 for layer 2-7, 16 for layer 8-13, and 7 for layer 14-19) yield higher accuracy (by 0.13%) and smaller bits/weights (by 0.13 bits/weights). As such, **FleXOR facilitates a fine-grained exploration of optimal quantization bit search** (given as fractional numbers determined by $N_{in}$, $N_{out}$, and $q$) that has not been available in the previous binary-coding-based quantization methods.

# 5 Experimental Results on ImageNet

In order to show that FleXOR principles can be extended to larger models, we choose ResNet-18 on ImageNet [21]. We use SGD optimizer with a momentum of 0.9 and an initial learning rate of 0.1. Batch size is 128, weight decay factor is $10^{-5}$, and $S_{tanh}$ is 10. Learning rate is reduced by half at the 70th, 100th, and 130th. For warmup, during the initial ten epochs, $S_{tanh}$ and learning rate increase linearly from 5 and 0.0, respectively, to initial values.

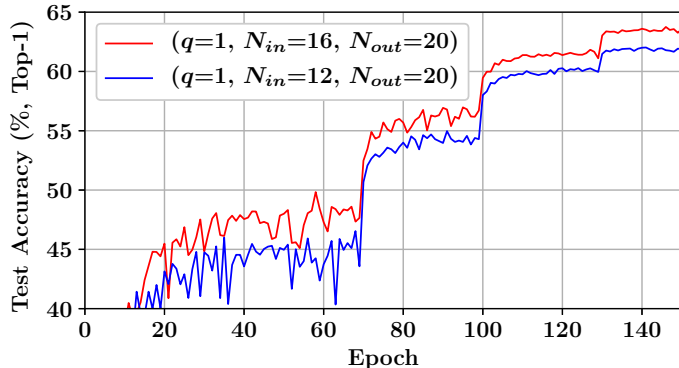

Figure 8: Test accuracy (Top-1) of ResNet-18 on ImageNet using FleXOR.

Figure 8 depicts the test accuracy of ResNet-18 on ImageNet when ($q$=1, $N_{in}$=16 and $N_{out}$=20) and ($q$=1, $N_{in}$=12 and $N_{out}$=20). Refer to Appendix for more results with $q$=2. Table 3 shows the comparison on model accuracy of ResNet-18 when weights are compressed by quantization (and additional encryption by FleXOR) while activations maintain full precision. Training ResNet-18 including FleXOR components is successfully performed. In Table 3, BinaryRelax and BWN do not modify the underlying model architecture, while ABC-Net introduces a new block structure of the convolution for quantized network designs. FleXOR achieves the best top-1 accuracy even with only 0.8bit/weight and demonstrates improved model accuracy as the number of bits per weight increases.

We acknowledge that there are numerous other methods to reduce the neural networks in size. For example, low-rank approximation and parameter pruning could be additionally performed to reduce the size further. We believe that such methods are orthogonal to our proposed method.

## 6 Conclusion

This paper proposes an encryption algorithm/architecture, FleXOR, as a framework to further compress quantized weights. Encryption is designed to produce more outputs than inputs by increasing the Hamming distance of output functions when output functions are linear functions of inputs. Output functions are implemented as a combination of XOR gates that are included in the model to find encrypted and quantized weights through gradient descent while using the tanh function for backward propagation. FleXOR enables fractional numbers of bits for weights and, thus, much wider trade-offs between weight storage and model accuracy. Experimental results show that ResNet on CIFAR-10 and ImageNet can be represented by sub 1-bit/weight compression with high accuracy.

## Broader Impact

Due to rapid advances in developing neural networks of higher model accuracy and increasingly complicated tasks to be supported, the size of DNNs is becoming exponentially larger. Our work facilitates the deployment of large DNN applications in various forms including mobile devices because of the powerful model compression ratio. As for positive perspectives, hence, a huge amount of energy consumption to run model inferences can be saved by our proposed quantization and encryption techniques. Also, a lot of computing systems that are based on binary neural network forms can improve model accuracy. We expect that lots of useful DNN models would be available for devices of low cost. On the other hand, some common concerns on DNNs such as privacy breaching and heavy surveillance can be worsened by DNN devices that are more available economically by using our proposed techniques.

## Acknowledgments

We would like to thank the anonymous reviewers for their valuable comments on our manuscript.

## Footnotes

[2]To 4 groups of 3×3 conv layers in ResNet-18 (except the first conv layer connected to the inputs), we assign 0.9, 0.8, 0.7, and 0.6 bits/weight, respectively. To the remaining 1×1 conv layers (performing downsampling), we assign 0.95, 0.9, and 0.8 bits/weight, respectively.

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
