[Supplementary Material]

# Appendix for
# "FleXOR: Trainable Fractional Quantization"

## A    Example of a XOR-gate Network Structure Representation

In Figure 2, outputs of an XOR-gate network are given as

$$y_1 = x_1 \oplus x_3 \oplus x_4$$
$$y_2 = x_1 \oplus x_2$$
$$y_3 = x_1 \oplus x_2 \oplus x_3$$
$$y_4 = x_3 \oplus x_4$$
$$y_5 = x_2 \oplus x_4$$
$$y_6 = x_2 \oplus x_3 \oplus x_4.$$

Equivalently, the same structure as above can be represented in a matrix as

$$\boldsymbol{M}^{\oplus} = \begin{bmatrix} 1 & 0 & 1 & 1 \\ 1 & 1 & 0 & 0 \\ 1 & 1 & 1 & 0 \\ 0 & 0 & 1 & 1 \\ 0 & 1 & 0 & 1 \\ 0 & 1 & 1 & 1 \end{bmatrix}. \tag{1}$$

Note that elements of $\boldsymbol{M}^{\oplus}$ are matched with coefficients of $y_i (1 \leq i \leq 6)$. For two vectors $\boldsymbol{y} = \{y_1, y_2, y_3, y_4, y_5, y_6\}$ and $\boldsymbol{x} = \{x_1, x_2, x_3, x_4\}$, the following equation holds:

$$\boldsymbol{y} = \boldsymbol{M}^{\oplus} \cdot \boldsymbol{x}, \tag{2}$$

where element-wise addition and multiplication are performed by 'XOR' and 'AND' function, respectively. In Eq. (1), $N_{tap}$ (i.e., the number of '1's in a row) is 2 or 3.

## B    Supplementary Data for Basic FleXOR Training Principles

A Boolean XOR gate can be modeled as $\mathcal{F}^{\oplus}(x_1, x_2) = (-1)\,\text{sign}(x_1)\,\text{sign}(x_2)$ if 0 is replaced with $-1$ as shown in Table 1.

| $\text{sign}(x_1)$ | $\text{sign}(x_2)$ | $\mathcal{F}^{\oplus}(x_1, x_2)$ |
|:---:|:---:|:---:|
| $-1$ | $-1$ | $-1$ |
| $-1$ | $+1$ | $+1$ |
| $+1$ | $-1$ | $+1$ |
| $+1$ | $+1$ | $-1$ |

Table 1: An XOR gate modeling using $\mathcal{F}^{\oplus}(x_1, x_2)$.

In Eq. (1), forward propagation for $y_3$ is expressed as

$$y_3 = \mathcal{F}^{\oplus}(x_1, x_2, x_3) = (-1)^2 \,\text{sign}(x_1)\,\text{sign}(x_2)\,\text{sign}(x_3). \tag{3}$$

while partial derivative of $y_3$ with respect to $x_1$ is given as (not derived from Eq. (3))

$$\frac{\partial y_3}{\partial x_1} = S_{\tanh}(-1)^2(1 - \tanh^2(x_1 \cdot S_{\tanh})) \tanh(x_2 \cdot S_{\tanh}) \tanh(x_3 \cdot S_{\tanh}), \qquad (4)$$

or as

$$\frac{\partial y_3}{\partial x_1} \approx S_{\tanh}(-1)^2(1 - \tanh^2(x_1 \cdot S_{\tanh})) \operatorname{sign}(x_2) \operatorname{sign}(x_3). \qquad (5)$$

We choose Eq. (5), instead of Eq. (4), as explained in Section 3.

Figure 1: The left graph shows hyperbolic tangent ($y = \tanh(x \cdot S_{\tanh})$) graphs with various scaling factors ($S_{\tanh}$), . The right graph shows their derivatives. These graphs support the arguments of '**Optimize $S_{\tanh}$**' in Section 4.

As shown in Figure 1, large $S_{\tanh}$ yields sharp transitions for near-zero inputs. Such a sharp approximation of the Heaviside step function produces large gradient values for small inputs and encourages encrypted weights to be separated into negative or positive values. Too large $S_{\tanh}$, however, has the same issues of a too-large learning rate.

Figure 4 presents training loss and test accuracy when $N_{tap}=2$ and $N_{out}$ is 10 or 20. Compared with Figure 5, $N_{tap}=2$ presents improved accuracy for the cases of high compression configurations (e.g., $N_{in}=4$ and $N_{out}=10$). We use $N_{tap} = 2$ for CIFAR-10 and ImageNet, since low $N_{tap}$ avoids gradient vanishing problems or high approximation errors in Eq.(5) or Eq.(6).

Figure 5 plots the distribution of encrypted weights at different training steps when each row of $\boldsymbol{M}^{\oplus}$ is randomly assigned with $\{0, 1\}$ (i.e., $N_{tap}$ is $N_{in}/2$ on average) or assigned with only two 1's ($N_{tap}=2$). Due to gradient calculations based on $\tanh$ and high $S_{\tanh}$, encrypted weights tend to be clustered on the left or right (near-zero encrypted weights become less as $N_{tap}$ increases) even without weight clipping.

## C  Supplementary Experimental Results of CIFAR-10 and ImageNet

In this section, we additionally provide various graphs and accuracy tables for ResNet models on CIFAR10 and ImageNet. We also present experimental results from wider hyper-parameters searches including $q=2$ with two separate $\boldsymbol{M}^{\oplus}$ configurations (with the same $N_{in}$ and $N_{out}$ for two $\boldsymbol{M}^{\oplus}$ matrices).

Figure 2: An example showing FleXOR operations for training. XOR gates are described in different ways for forward- and backward propagation. Once we obtain encrypted binary weights after training, we use digital XOR gates for inference.

Figure 3: Using the same weight storage footprint, FleXOR enables various internal quantization schemes. (Left): 1-bit internal quantization. (Right): 3-bit internal quantization with 3 different $M^{\oplus}$ configurations.

| | Bits/Weight | ResNet-20 | | ResNet-32 | | Comp. Ratio |
|---|---|---|---|---|---|---|
| FP | 32 | 91.87% | Diff. | 92.33% | Diff. | 1.0x |
| $N_{in}$=10, $N_{out}$=10 | 1.0 | 90.21% | -1.66 | 91.40% | -0.93 | 29.95$\times$ |
| $N_{in}$=9, $N_{out}$=10 | 0.9 | 90.03% | -1.84 | 91.28% | -1.05 | 31.82$\times$ |
| $N_{in}$=8, $N_{out}$=10 | 0.8 | 89.73% | -2.14 | 90.96% | -1.37 | 35.32$\times$ |
| $N_{in}$=7, $N_{out}$=10 | 0.7 | 89.88% | -1.99 | 90.67% | -1.66 | 39.68$\times$ |
| $N_{in}$=6, $N_{out}$=10 | 0.6 | 89.21% | -2.66 | 90.41% | -1.92 | 45.27$\times$ |
| $N_{in}$=5, $N_{out}$=10 | 0.5 | 88.59% | -3.28 | 89.95% | -2.38 | 52.70$\times$ |

Table 2: Weight compression comparison of ResNet-20 and ResNet-32 on CIFAR-10 when $N_{out}$=10. Parameters and recipes not described in the table are the same as in Table 1. We also present compression ratio for fractional quantized ResNet-32 when one scaling factor ($\alpha$) is assigned to each output channel.

Figure 4: Test accuracy and training loss of LeNet-5 on MNIST when number of '1's in each row of $M^{\oplus}$ is fixed to be 2 ($N_{tap}$=2). $N_{out}$ is 10 or 20 to generate, effectively, 0.4, 0.6, or 0.8 bit/weight quantization. With low $N_{tap}$ of $M^{\oplus}$, MNIST training presents less variations on training loss and test accuracy that in Figure 5.

Figure 5: Distribution of encrypted weight values for FC1 layer of LeNet-5 at different training steps using $S_{\tanh}$=100 and $N_{out}$=10. (Left): $M^{\oplus}$ is randomly filled ($N_{tap} \approx N_{in}/2$). (Right): $N_{tap} = 2$ for every row of $M^{\oplus}$.

| | ResNet-20 | | | ResNet-32 | | |
|---|---|---|---|---|---|---|
| | FP | Quant. | Diff. | FP | Quant. | Diff. |
| TWN (ternary) | 92.68% | 88.65% | -4.03 | 93.40% | 90.94% | -2.46 |
| BinaryRelax (ternary) | 92.68% | 90.07% | -1.91 | 93.40% | 92.04% | -1.36 |
| TTQ (ternary) | 91.77% | 91.13% | -0.64 | 92.33% | 92.37% | +0.04 |
| LQ-Net (2 bit) | 92.10% | 91.80% | -0.30 | - | - | - |
| FleXOR($q = 2, N_{out} = 20$) | | | | | | |
| $N_{in}$=20, 2.0 bit/weight | | 91.38% | -0.49 | | 92.25% | -0.08 |
| $N_{in}$=18, 1.8 bit/weight | | 91.00% | -0.87 | | 92.27% | -0.06 |
| $N_{in}$=16, 1.6 bit/weight | 91.87% | 90.88% | -0.99 | 92.33% | 92.11% | -0.22 |
| $N_{in}$=14, 1.4 bit/weight | | 90.90% | -0.97 | | 92.02% | -0.31 |
| $N_{in}$=12, 1.2 bit/weight | | 90.56% | -1.31 | | 91.62% | -0.71 |
| FleXOR($q = 2, N_{out} = 10$) | | | | | | |
| $N_{in}$=10, 2.0 bit/weight | | 91.19% | -0.68 | | 92.61% | +0.28 |
| $N_{in}$=9, 1.8 bit/weight | | 91.44% | -0.43 | | 92.09% | -0.24 |
| $N_{in}$=8, 1.6 bit/weight | 91.87% | 91.10% | -0.77 | 92.33% | 92.08% | -0.25 |
| $N_{in}$=7, 1.4 bit/weight | | 90.94% | -0.93 | | 91.74% | -0.59 |
| $N_{in}$=6, 1.2 bit/weight | | 90.56% | -1.31 | | 91.37% | -0.96 |

Table 3: Weight compression comparison of ResNet-20 and ResNet-32 on CIFAR-10 using learning rate warmup (for 100 epochs) and $q$=2. As mentioned in Figure 6, multiple $M^{\oplus}$ can be combined for multi-bit quantization schemes. Then, the number of scaling factors should be doubled. FleXOR with $q$=2 and two different $M^{\oplus}$ structures achieve full-precision accuracy when both $N_{in}$ and $N_{out}$ are 10.

(a) First convolution layer in Layer1

(b) Last convolution layer in Layer1

(c) First convolution layer in Layer2

(d) Last convolution layer in Layer2

Figure 6: Distributions of encrypted weights (at the end of training) in various layers of ResNet-32 on CIFAR-10 using various $S_{\mathrm{tanh}}$ and the same $N_{out}$, $N_{in}$, and $q$ as Figure 7. The ResNet-32 network mainly consists of three layers according to the feature map sizes: Layer1, Layer2 and Layer3.

| Methods | Bits/Weight | Top-1 | Top-5 |
|---|---|---|---|
| Full Precision [1] | 32 | 69.6% | 89.2% |
| TWN [3] | ternary | 61.8% | 84.2% |
| ABC-Net [4] | 2 | 63.7% | 85.2% |
| BinaryRelax [5] | ternary | 66.5% | 87.3% |
| TTQ(1.5× Wide) [7] | ternary | 66.6% | 87.2% |
| LQ-net [6] | 2 | 68.0% | 88.0% |
| QIL [2] | 2 | 68.1% | 88.3% |
| FleXOR ($q$=2, $N_{out}$=20) | 1.6 (0.8×2) | 66.2% | 86.7% |
| | 1.2 (0.6×2) | 65.4% | 86.0% |
| | 0.8 (0.4×2) | 63.8% | 85.0% |

Table 4: Weight compression comparison of ResNet-18 on ImageNet when $q$=2. Since $q$ is 2, we also list the other compression schemes which use 2-bit or ternary quantization scheme for model compression.

(a) **Initial Learning Rate (0.1)**: Test accuracy of ResNet-32 on CIFAR10 using the learning schedule in Figure 7 and various initial learning rates (0.05, 0.1, 0.2, 0.5).

(b) **No Weight Clipping**: Test accuracy of ResNet-32 on CIFAR10 using the learning schedule in Figure 7. As for weight clipping, we restrict the encrypted weights to be ranged as $(-2.0/S_{\text{tanh}}, +2.0/S_{\text{tanh}})$. As can be observed, the red line implies that weight clipping is not effective with FleXOR.

(c) **Weight Decay Factor ($10^{-5}$)**: Two graphs depict test accuracy of ResNet-18 on ImageNet with or without weight decay. The learning rate in the red line (no weight decay) is reduced by half at the $100^{\text{th}}$, $130^{\text{th}}$ and $150^{\text{th}}$ epochs. The learning rate of the blue line (with weight decay) is reduced by half at $70^{\text{th}}$, $100^{\text{th}}$ and $130^{\text{th}}$ epochs. With weight decay (blue graph), despite slow convergence in the early training steps, model accuracy is eventually higher than the red one without weight decay scheme.

Figure 7: Comparison of various hyper-parameter choices for CIFAR-10 or ImageNet.

(a) Test accuracy using $q=1$.

(b) Test accuracy using $q=2$. Compared to the above plots (Figure 8a), this figure shows that a combination of multiple $M^{\oplus}$ for a binary code can lead to stable learning curves and higher model accuracy.

Figure 8: Test accuracy of ResNet-32 on CIFAR-10 using learning rate warmup (for 100 epochs) and $N_{out}=20$