[Reviews · NeurIPS 2020]

Review 1

Summary and Contributions: The authors propose to quantize the weights of a neural network by enabling a fractional number of bits per weight. They use a network of differentiable XOR gates that maps encrypted weights to higher-dimensional decrypted weights to decode the parameters on-the-fly and learn both the encrypted weights and the scaling factors involved in the XOR networks by gradient descent.

Strengths: The use of XOR gates to improve quantization seems interesting. The authors use a network of differentiable XOR gates that maps encrypted weights to higher-dimensional decrypted weights to decode the parameters on-the-fly and learn both the encrypted weights and the scaling factors involved in the XOR networks by gradient descent.

Weaknesses: More extensive and thorough experiments are needed. The authors proposed a task-independent quantization method thus they must provide experiments based on different tasks like image detection using Mask R-CNN. The sub-bit quantization achieved by this method should be interesting to the industrial. However, the authors must show how this quantization affects challenging deep networks like ResNet-50 architecture. What happens to the average ratio in the case were different layers of the same network are quantized with distinct integer ratio of bits per weights? Is fractional or not? Is this a problem? ======================= Post-rebuttal update: Thank the authors very much for detailed responses. I have also read other reviews, and I am fine with accepting this paper. I am expecting the authors to include the answers in the text (reporting the limitations of their method is important).

Correctness: The claims and method are correct.

Clarity: The paper is very well written and completed with nice visualizations and pseudocode.

Relation to Prior Work: Yes, it Is clearly discussed how this work differs from previous contributions.

Reproducibility: Yes

Additional Feedback:


Review 2

Summary and Contributions: The main motivation of this paper is to provide a method to explore the potential of using less than 1 bit to represent neural network weights. The paper proposes a sound method of not only flexibly apportioning representational power to weights, but also a method to make that apportioning learnable during backpropagation. The contributions include the complete method (which could be applied to any neural network topology) to train and infer with fractional bit representations and showing that different layers in an NN are more sensitive to this representational degradation than others. To me, the main contribution is the existence proof that gradual degradation in performance occurs when representation per weight is reduced <1b, <0.8b, etc. This is important for future directions as it hints at the ability to squeeze more representational power out of intrinsically binary hardware (digital CMOS being the dominant paradigm).

Strengths: The paper proposes a clear hypothesis that neural networks can train at less than 1 bit of precision per weight. They show a method that demonstrates this is possible and propose a sound method to do so. The proposed solution to map a smaller number of bits to a larger number of bits to represent weights also happens to be differentiable and hardware friendly, making it plausible in real implementations. The authors found that their method can achieve similar performance to other state-of-the-art single bit methods while compressing the representation of the weights. A major contribution is to show that a learned apportionment of bits across multiple weights is possible and could lead to new ways of maximizing the total representational power of any binary computer.

Weaknesses: I’d like to have seen a bit of an explanation of why this works. Is it that every weight utilizes less than 1b of representation? Or do some weights use >1b while others can use much less? Was the reason for decent performance of the trained network due to the ability to apportion representation or does that not matter? Some kind of information theoretic analysis could add a lot of depth to this paper. The paper doesn’t discuss any other possible methods for accomplishing a similar thing. For instance, is this method really required or does the fact that fractional bits work an indication that the neural net is too overspecified? Could the neural network be reduced in size and accomplish the same thing?

Correctness: The methodology of proposing a new numerical method, testing it against established network topologies and data, and reporting classification rates is pretty standard. The authors also do a good job of comparing to other published methods. I don’t see anything incorrect in the encryption scheme or its inverse (used for backpropagation). The experimental results, however, are a bit harder to generalize. I would like to see this in other flavors of NNs other than ResNets (ConvNets) and see if the same principles apply to Transformers or other topologies

Clarity: The paper has several instances of discussing a topic without context. In fact, the first sentence in the abstract, lines 1-3, don’t have enough context for the reader. A succinct summary of the problem is the right way to start. Also, the term “binary codes” can have many different interpretations. It is not immediately clear what interpretation this paper is referring to. Line 44: what is meant by “different bits”? Is it meant to be “different number of bits”? Line 103-104: Is this implying there is a tradeoff between memory and compute requirements? Just say that. Line 108: The jump from eq 5 to simplified eq 6 wasn’t clear. Why does this work? Expand on this jump. The authors begin to use terms in section 4 that were not previously defined: Line 144. Ntap isn’t defined until later in this paragraph, but is used in the bolded text. Please define these terms before using them. Other readers don’t have the same familiarity with your work. In figure 7 and Line 175: I could never find a definition of ‘q’. Again, please define these things before using them. Line 169: What was the methodology for finding Stanh in this work?

Relation to Prior Work: The authors discuss and cite other methods used for binary NNs. They do a good job relating this to other work around very low precision for neural networks by comparing classification results. However, given that the work borrows from other fields (communications and compression), more discussions and citations towards those other fields’ work would really strengthen the paper. For instance, flexibly trading off information redundancy for message size is common in communication networks. Concepts like coding gain specifically analyse the optimal tradeoffs and techniques like Viterbi encoding or puncture encoding are well-analysed in that literature. This work is doing something very similar, albeit, with a different application.

Reproducibility: Yes

Additional Feedback: Code examples could go a long way here. I acknowledge any rebuttals and stand by my review. The current review includes any updates based on rebuttals.


Review 3

Summary and Contributions: The authors proposed a new method, FleXOR, to decompress weight values used by convolutions from a compact representation by using a small Boolean function, XOR-gate network. The input to the Boolean function is a compact chunk of binary values, and the output is a binary vector, when multiplied with the layer-wise vector of scaling factors, it gives the individual weight values. The XOR-gate network is completely binarized in the forward pass, and in backpropagation the sign functions are approximated with scaled tanh functions. This allows the binarized network to be trained with SGD. They found their results are competitive when compared to the state-of-the-art, with accuracy improvements under the same compression ratio.

Strengths: The idea of designing a trainable binarized network that deflates encoded weights is interesting, and the reviewer believes it is also novel. FleXOR also shows promising results in improving the accuracy-compression trade-off. It is also reasonable to believe that the added latency and area of FleXOR is minimal in custom hardware. This paper has a good amount of ≤1-bit per weight experimental results. The reviewer also appreciates the detailed experimental setup in Section 4 that is helpful for reproducing the results reported in the paper.

Weaknesses: The authors proposed in Figure 1 to use binarized inputs to allow the convolution to be fully bit-wise. The reviewer did not find results in the paper that used quantized inputs. In Algorithm 1, the choice of where to apply the XOR-gate function is eccentric. It seems to be oblivious about the shape of input and output tensors and considers chunks of flattened tensors. Encoded weights risked inference between consecutive filters, but fortunately the end results seem to be good. In both Tables 1 and 3, the authors reported 1-bit weight with full-precision activation results from other methods. The reviewer believes that this is not sufficient, and unfair to other state-of-the-art methods. For instance, for both LQ-Net and ABC-Net, with 1-bit weight and full-precision activation, the weights are simply binarized to {-1, 1} with a scaling. This is a degenerate case that do not leverage binary codes. As the reviewer explains below in Additional feedback, other work reported substantially higher top-1 accuracies with 2 or 3-bit weight quantizations and full precision activations.

Correctness: It is reasonable to believe the correctness of the method and results used in this paper. The authors provided sufficient details in the description of the methods and hyperparameters used for reproducing the work. Some of the experiments were also repeated to give error bars on the results.

Clarity: The paper is overall easy to follow, with minor grammatical errors: * Line 24 mentions that α is a scaling factor with `α ∈ R` but should instead be a vector.

Relation to Prior Work: The paper clearly differentiates their contribution from the previous work. FleXOR is designed as an additional layer on top of the binary-code based quantization methods.

Reproducibility: Yes

Additional Feedback: It would noticeably improve this paper if the authors could report higher-bit results (2 and 3 bits) to fill out the accuracy-compression trade-off. This would allow many more state-of-the-art methods to be compared against. Other publications produced ResNet-18+ImageNet results that are substantially better (66%~69% top-1 accuracies) with 10x~16x compression (2-3 bits) [1-6]. The authors could also report other comparative methods that are simple to implement. For example, Huffman coding instead of FleXOR. [1]: Chenzhuo Zhu, et al. Trained ternary quantization. ICLR 2017. [2]: Aojun Zhou, et al. Incremental network quantization: Towards lossless CNNs with low-precision weights. ICLR 2017. [3]: Cong Leng, et al. Extremely low bit neural network: Squeeze the last bit out with ADMM. AAAI 2018. [4]: Dongqing Zhang, et al. LQ-Nets: Learned quantization for highly accurate and compact deep neural networks. ECCV 2018. [5]: Abhimanyu Dubey, et al. Coreset-based neural network compression. ECCV 2018. [6]: Yiren Zhao, et al. Focused Quantization for Sparse CNNs. NeurIPS 2019. Updates after the rebuttal ------------------------ R3Q1: Thanks. The reviewer would also like to point out that if you are not using binary-coded activations, then it is not clear how you can have "bit-wise computation" for the "computation scheme". R3Q2: The reviewer understands the authors' intention of using FleXOR in a custom accelerator. The original question raised by the reviewer were related to how it impacts accuracy rather than computational efficiency. As it is clear from Algorithm 1 the unencoded binary codes for weights are generated in chunks that do not align with the shape of the weight tensor used for convolution. The misalignment may introduce interference between filters and subsequently impact accuracy. It would be great if this can addressed in the paper. R3Q3: It concerns the reviewer as the paper only compares q=1 to SOTA. The reviewer understands that the authors aim for the best sub-1-bit method, but it would improve the paper substantially by comparing FleXOR using additional quantization levels (q > 1) with other related SOTAs. The reviewer would also appriecate if the paper can also report other comparative methods that are simple to implement. For example, Huffman coding instead of FleXOR.


Review 4

Summary and Contributions: The paper proposes to compress quantized weights with an XOR-based encryption network so that the bit-depth can go below 1 while the network is still end-to-end trainable.

Strengths: Novelty: This is the first attempt to push quantization precision to below 1 bit. Empirical evaluation: the list of training techniques paired with ablation studies help to understand the sensitivity of the proposed method.

Weaknesses: Significance: On ImageNet, it can achieve better memory saving with similar top-1 accuracy as previous SOTA, but the gain is marginal: (Table 3) 63.8%@40x for FleXOR with 0.8 bits vs DSQ's 63.7%@32x. Clarity / significance: It wasn't clear to me how the XOR-gate network (i.e. M^+ in line 72) is structured? If the design is manual and has to be tailored for each problem, then it would undermine the significance of this work; if it's learned / fixed, then it'd best to show theoretical analysis of its compression factor in relation to the theoretical optimal. Please answer this in the rebuttal.

Correctness: Yes.

Clarity: I'm not super clear upon first reading of how the XOR-gate network is structured.

Relation to Prior Work: Yes.

Reproducibility: Yes

Additional Feedback:

[Author Response · NeurIPS 2020]

**(R1Q1)** More extensive and thorough experiments are needed.

(A) Since our major contribution is to enable fractional sub 1-bit weight quantization, we selected models that have
been previously quantized by '1' bit/weight for comparisons on the model accuracy. As a result, unfortunately, the
range of model selections is somewhat limited as we present in Table 1 and 3.

**(R1Q2)** What about average ratio when different layers are quantized with distinct integer ratio of bits per weights?

(A) We acknowledge that fractional numbers can be obtained when considering the average number of quantization bits.
The number of quantization bits per layer is, however, still limited to be an integer such that the minimum number of
quantization bits is '1' while we proved that some layers can be quantized with 0.35 bits/weight without noticeable
accuracy degradation (as shown in Table 2). Sub 1-bit quantization is only available through FleXOR.

**(R2Q1)** Every weight utilizes less than 1b? Or do some weights use >1b while other can use much less?

(A) It would be an exciting research topic to study the distribution of the optimal number of quantization bits for each
weight. We believe that such distribution would be wide and some weights require >1b while numerous weights need
<1b because 1) increasing $N_{in}$ and $N_{out}$ allows such distributions to be wider and enhances model accuracy even for
the same compression ratio and 2) as shown in Table 1, model accuracy of 1-bit quantization with FleXOR is higher
than other quantization schemes that do not include encoding schemes. We will add related discussions with supporting
experiments in the final manuscript.

**(R2Q2)** The paper doesn't discuss any other possible methods for accomplishing a similar thing.

(A) We understand that there are numerous methods to reduce the neural networks in size. For example, low-rank
approximation and parameter pruning could be additionally performed to reduce the size further. We believe that such
methods are orthogonal to our proposed method while we feel that our contributions in this work are 1) rethinking
the limits of quantization method through encoding schemes and 2) providing a new compression method using the
techniques that have been widely studied in digital communication or cryptography.

**(R2Q3)** Issues on clarity

(A) We appreciate your careful reviews. We will revise the abstact, correct typos, and introduce some definitions clearly
in the revised manuscript.

**(R3Q1)** Figure 1 uses binarized inputs. The reviewer did not find results in the paper that used quantized inputs.

(A) We will fix Figure 1 since inputs are not quantized. "Input weight format" should read "Internal weight format."

**(R3Q2)** Algorithm 1 considers flattened tensors. Encoded weights risked inference between consecutive filters.

(A) Algorithm 1 describes hardware operations (that are best implemented by ASIC or FPGA) rather than instructions to
be operated by CPUs or GPUs. We acknowledge that Algorithm 1 may lead to significant overhead when implemented
by general-purpose computing systems. We will add discussions to address your concern.

**(R3Q3)** In both Tables 1 and 3, the authors reported 1-bit weight with full-precision activation results from other
methods. The reviewer believes that this is not sufficient.

(A) We want to point out that the goal of our work is not to present the best model accuracy with 2-3 quantization bits.
Previously, it has been known that the form of binary neural networks is the minimum for quantization. On the other
hand, throughout this work, we show that fractional sub 1-bit quantization is possible by using XOR gates. Accordingly,
we selected models that have been quantized by 1-bit per weight while we could not quantize activations because once
activations are quantized, model accuracy is affected by the choice of activation quantization method. Activations can
be quantized additionally in our work.

**(R4Q1)** FleXOR can achieve better memory saving with similar accuracy as previous SOTA, but the gain is marginal.

(A) Outperfoming previous models even with sub 1-bit quantization can be important but not our major target in this
work. As we include the results with 0.63 or 0.6 bits/weight in Table 3, our goal is to demonstrate that even sub
1-bit quantization is enabled by FleXOR with graceful accuracy degradation. We believe that FleXOR presents a new
inference design paradigm with much wider trade-off search space between model accuracy and compression ratio.

**(R4Q2)** Clarify how to structure XOR-gate networks.

Since an XOR-gate network is shared by many weights (such as 1 million), it is difficult (if not impossible) to manually
optimize an XOR-gate network. Hence, random network configuration is enough to fulfill the purpose of random
number generation. In short, XOR-gate network design is simple and straightforward. We will add discussions to
address your concern in the revised manuscript.

[Meta-Review · NeurIPS 2020]

There is a consensus from all parties that this is a valuable contribution to the field. However I do emphasise that it is very important that the authors rework their paper to incorporate the answers in their rebuttal into the paper itself. They would significantly enhance the paper, and the reviewers have specifically asked me to say that they think this is vital.